# G_1_ Cell Cycle Arrest and Extrinsic Apoptotic Mechanisms Underlying the Anti-Leukemic Activity of CDK7 Inhibitor BS-181

**DOI:** 10.3390/cancers12123845

**Published:** 2020-12-19

**Authors:** Shin Young Park, Ki Yun Kim, Do Youn Jun, Su-Kyeong Hwang, Young Ho Kim

**Affiliations:** 1Laboratory of Immunobiology, School of Life Sciences, BK21 FOUR KNU Creative BioResearch Group, Kyungpook National University, Daegu 41566, Korea; psy0419@knu.ac.kr (S.Y.P.); kky@knu.ac.kr (K.Y.K.); 2Astrogen Inc., Techno-Building 313, Kyungpook National University, Daegu 41566, Korea; dyjun@knu.ac.kr (D.Y.J.); skhwang@knu.ac.kr (S.-K.H.); 3Department of Pediatrics, School of Medicine, Kyungpook National University, Daegu 41566, Korea

**Keywords:** CDK inhibitor, G_1_ arrest, extrinsic apoptotic pathway, TRAIL, DR5, leukemia

## Abstract

**Simple Summary:**

Chemotherapy resistance in human T-cell acute lymphoblastic leukemia (T-ALL), an aggressive neoplasm, results in poor prognosis despite advances in treatment modalities. Toward the identification of an effective alternative, in the present study, we elucidated the mechanism underlying the antitumor activity of the CDK7 inhibitor BS-181 using malignant cells (Jurkat A3, U937, and HeLa) and normal human peripheral T cells. This is the first report to demonstrate that BS-181 antitumor activity is mainly caused by extrinsic apoptosis induction through cell-surface TRAIL/DR5 levels in human T-ALL Jurkat T cells. Moreover, combined treatment with recombinant TRAIL (rTRAIL) exerted synergistic effects on BS-181 cytotoxicity against malignant cells but not normal human peripheral T cells by augmenting both the extrinsic and intrinsic BCL-2-sensitive apoptosis pathways. Our findings suggest that the combination with rTRAIL may facilitate BS-181 antitumor activity against T-ALL cells while minimizing associated side effects, therefore potentially being applicable to clinical human T-ALL treatment.

**Abstract:**

In vitro antitumor activity of the CDK7 inhibitor BS-181 against human T-ALL Jurkat cells was determined. Treatment of Jurkat clones (JT/Neo) with BS-181 caused cytotoxicity and several apoptotic events, including TRAIL/DR4/DR5 upregulation, c-FLIP down-regulation, BID cleavage, BAK activation, ΔΨ_m_ loss, caspase-8/9/3 activation, and PARP cleavage. However, the BCL-2-overexpressing Jurkat clone (JT/BCL-2) abrogated these apoptotic responses. CDK7 catalyzed the activating phosphorylation of CDK1 (Thr161) and CDK2 (Thr160), and CDK-directed retinoblastoma phosphorylation was attenuated in both BS-181-treated Jurkat clones, whereas only JT/BCL-2 cells exhibited G_1_ cell cycle arrest. The G_1_-blocker hydroxyurea augmented BS-181-induced apoptosis by enhancing TRAIL/DR4/DR5 upregulation and c-FLIP down-regulation. BS-181-induced FITC–annexin V-positive apoptotic cells were mostly in the sub-G_1_ and G_1_ phases. BS-181-induced cytotoxicity and mitochondrial apoptotic events (BAK activation/ΔΨ_m_ loss/caspase-9 activation) in Jurkat clones I2.1 (FADD-deficient) and I9.2 (caspase-8-deficient) were significantly lower than in A3 (wild-type). Exogenously added recombinant TRAIL (rTRAIL) markedly synergized BS-181-induced apoptosis in A3 cells but not in normal peripheral T cells. The cotreatment cytotoxicity was significantly reduced by the DR5-blocking antibody but not by the DR4-blocking antibody. These results demonstrated that the BS-181 anti-leukemic activity is attributed to extrinsic TRAIL/DR5-dependent apoptosis preferentially induced in G_1_-arrested cells, and that BS-181 and rTRAIL in combination may hold promise for T-ALL treatment.

## 1. Introduction

Human T-cell acute lymphoblastic leukemia (T-ALL) is an aggressive neoplasm derived from the malignant transformation of lymphoblasts committed to a T-cell lineage and accounts for approximately 20% of all ALL cases [1,2]_._ Although recent advances in risk-adapted chemotherapy regimens improve the overall survival rates in both childhood and adult T-ALL, resistance to chemotherapy and early relapse can occur in T-ALL, with high-risk features leading to unfavorable prognosis and low survival rates [3,4]. Thus, to improve the overall survival rates in the chemotherapeutic treatment of T-ALL and other high-risk leukemias, there is a high demand for novel antitumor agents that can minimize drug resistance and side effects [5].

Cell cycle progression is positively controlled by a family of cyclin-dependent kinases (CDKs), which includes three interphase CDKs (CDK2, CDK4, and CDK6), one mitotic CDK (CDK1), a regulatory CDK (CDK7, the catalytic subunit of the CDK-activating kinase (CAK) complex), and transcriptional CDKs (CDK8 and CDK9) [5,6,7,8]. Notably, the catalytic activities of CDKs are frequently elevated in conjunction with the uncontrolled proliferation of tumor cells, owing to genetic and epigenetic alterations in CDKs or their regulatory proteins, such as CDK inhibitors and cyclins [9,10,11,12,13]. Thus, CDKs have emerged as attractive targets for the development of antitumor agents and chemotherapeutic treatment [8,9,14]. Although first-generation CDK inhibitors with broad specificity showed low efficacy in clinical trials, second-generation CDK inhibitors targeting a narrow spectrum of CDKs yielded better clinical outcomes. In particular, recent advances in the development of highly specific CDK4/6 inhibitors have led to the approval of palbociclib and ribociclib for breast tumor treatment [15,16,17]. Along with attempts to expand pharmacological target CDKs to non-canonical CDKs, such as CDK7 to CDK9, further efforts have been made to develop third-generation CDK inhibitors with greater specificity and potency against their targets.

Compared with other CDK family members, the role of CDK7 in the cell cycle is unique, in that CDK7, together with cyclin H and MAT1, forms the CAK complex, which is responsible for the activating phosphorylation of CDK1, CDK2, CDK4, and CDK6 [18]. Moreover, the CAK complex exists as a part of the transcription factor TFIIH and phosphorylates the C-terminal domain of the largest subunit of RNA polymerase II (Pol II) for the transcriptional initiation of RNA Pol II-directed genes. In addition, CAK or TFIIH-associated CAK also phosphorylates several other transcription factors to regulate their activities [19,20]. These previous data suggest that CDK7, as an essential regulator of cell cycle progression and transcription, may constitute a critical therapeutic target for tumors. Recently, a pyrazolo [1,5-α] pyrimidine-derived compound, BS-181, which selectively inhibits CDK7 (the catalytic subunit of CAK) activity at low nanomolar concentrations in vitro, has been developed as a third-generation CDK inhibitor via a structure-based, computer-aided drug design [21]. The cytotoxic effect of BS-181 on solid tumors, including gastric tumor cells, was reported to be associated with G_1_ cell cycle arrest and mitochondria-dependent apoptotic cell death [21,22]. However, to clarify the antitumor activity of BS-181, further investigation regarding the correlation between these effects is required. Moreover, little is known regarding the involvement of the extrinsic apoptotic pathway in BS-181-induced apoptosis or whether the antitumor activity of BS-181 might be applied to T-ALL and other leukemias.

In this study, to elucidate a relationship between BS-181-induced G_1_ cell cycle arrest and mitochondrial apoptosis pathway activation, the effect of BS-181 on Jurkat T cell clones stably transfected with an empty vector (JT/Neo) or a vector expressing the anti-apoptotic factor B-cell lymphoma 2 (JT/BCL-2) was investigated. To examine the involvement of the extrinsic pathway in BS-181-induced apoptosis, apoptotic responses were compared not only in BS-181-treated Jurkat A3 (wild-type), I2.1 (FADD-deficient), and I9.2 (caspase-8-deficient) cells, but also in A3 cells cotreated either with recombinant human tumor necrosis factor-related apoptosis-inducing ligand (rTRAIL) or with neutralizing anti-death receptor 5 (DR5) antibody. The half-maximal inhibitory concentration (IC_50_) values of BS-181 in human acute leukemia cell lines (Jurkat A3 and U937) and human cervical carcinoma cell line (HeLa) were compared to those in normal human peripheral T cells in the absence or presence of exogenous rTRAIL to examine whether selective BS-181 cytotoxicity toward malignant tumor cells might be enhanced by the combined treatment with rTRAIL.

## 2. Results

### 2.1. BS-181 Cytotoxicity Is Exerted by G_1_ Cell Cycle Arrest and BCL-2-Sensitive Apoptosis in Jurkat T Cells

To examine whether BCL-2-sensitive mitochondrial apoptosis is a prerequisite for BS-181 cytotoxicity (Figure 1a), its cytotoxic effects on JT/Neo and JT/BCL-2 cells were compared. The viabilities of JT/Neo cells exposed to 5, 10, 15, and 20 μM BS-181 for 20 h were 97.3%, 77.5%, 54.8%, and 20.4%, respectively, with an IC_50_ value of 14.4 μM, whereas those of JT/BCL-2 cells overexpressing BCL-2 were 99.5%, 92.5%, 86.0.0%, and 84.0%, respectively (Figure 1b). Similarly, treatment with BS-181 at concentrations of 10 and 15 µM for 20 h increased the apoptotic sub-G_1_ peak to 23.2% and 54.8%, respectively, in JT/Neo cells, whereas no effect on this peak was observed in JT/BCL-2 cells (Figure 1c,d). Under these conditions, BS-181-induced G_1_ cell cycle arrest was observed only in JT/BCL-2 cells. BS-181-induced G_1_ arrest was also observed in Jurkat T cells overexpressing anti-apoptotic BCL-XL (Appendix A).

Fluorescein isothiocyanate (FITC)–annexin V and propidium iodide (PI) staining of JT/Neo cells treated with 15 µM BS-181 for 20 h showed that early apoptotic cells (stained only with FITC–annexin V) and late apoptotic cells (stained with both FITC–annexin V and PI) increased to 38.0% and 9.1%, respectively; however, necrotic cells (stained with only PI) were negligible (Figure 1e,f). When the forward scatter (FSC) distributions of unstained, early apoptotic, and late apoptotic cells were compared in BS-181-treated JT/Neo cells, both early and late apoptotic JT/Neo cells showed a reduction in cell size, indicative of typical apoptotic cellular shrinkage rather than necrotic cellular swelling. However, a BS-181-induced increase in the number of early and late apoptotic cells was not observed in JT/BCL-2 cells. These results suggested that BS-181 cytotoxicity toward Jurkat T cells was attributed partly to a cytostatic effect exerted by inducing G_1_ cell cycle arrest and mainly to cell death resulting from BCL-2-sensitive apoptosis induction, which might be preferentially induced in G_1_ phase cells. These results also suggested that G_1_ arrest was induced by a mechanism independent of the anti-apoptotic activity of BCL-2.

### 2.2. BCL-2 Overexpression Abrogates Extrinsic TRAIL/DR5 Upregulation-Mediated Apoptosis and Subsequent Mitochondrial Damage-Mediated Apoptosis in BS-181-Treated JT/Neo Cells

To examine the involvement of BCL-2-sensitive mitochondrial damage in BS-181-induced apoptosis, mitochondrial membrane potential (ΔΨ_m_) loss of BS-181-treated JT/Neo and JT/BCL-2 cells was analyzed by flow cytometry using 3,3′-dihexyloxacarbocyanine iodide (DiOC_6_) staining. The negative fluorescence percentages in JT/Neo cells treated with BS-181 at concentrations of 10 and 15 µM were 27.0% and 66.0%, respectively; however, BS-181-induced ΔΨ_m_ loss was abrogated in JT/BCL-2 cells (Figure 2a,b). After treatment with 10 and 15 μM BS-181, the BAK activation rates were 25.1% and 58.1% in JT/Neo cells, respectively, but negligible in JT/BCL-2 cells (Figure 2c,d). Under these conditions, BCL-2, BCL-XL, and MCL-1 levels remained relatively constant in BS-181-treated JT/Neo cells (Figure 2e). Additionally, neither the electrophoretic mobility reduction of BIM during sodium dodecyl sulfate (SDS)–polyacrylamide gel electrophoresis, which would indicate its activation [23], nor an elevation of BAK levels, was detected. Consistent with previous studies showing that BAK activation and resultant ΔΨ_m_ loss precede mitochondrial cytochrome *c* release into the cytosol, resulting in caspase-9/caspase-3 activation [24,25,26], BAK activation, ΔΨ_m_ loss, caspase-9/3 activation, and poly(ADP-ribose) polymerase (PARP) cleavage were detected in BS-181-treated JT/Neo cells, but not in JT/BCL-2 cells. These results demonstrated that the intrinsic BAK-dependent mitochondrial apoptosis pathway, which could be blocked by anti-apoptotic BCL-2, was involved in BS-181-induced apoptosis of Jurkat T cells.

As the extrinsic apoptosis pathway triggered by the upregulation of DR and/or TRAIL levels and down-regulation of c-FLIP can lead to BAK activation through the action of truncated BID (tBID, 15 kDa) produced from cleavage of BID (22 kDa) by active caspase-8 [27,28,29,30], the expression levels of the components of the extrinsic apoptosis pathway were investigated in BS-181-treated JT/Neo and JT/BCL-2 cells. Following the BS-181-induced elevation of the total cellular levels of death receptor 4 (DR4)/DR5 and TRAIL, caspase-8 activation and a decreased level of BID—reflecting its cleavage into tBID—were observed in BS-181-treated JT/Neo cells (Figure 2f). A significant decrease in the c-FLIP_L_ level, which acts as an extrinsic apoptosis inhibitor via blocking DR-mediated caspase-8 activation by preventing the binding of pro-caspase-8 to FADD [29,30], was detected in JT/Neo cells. Secreted soluble TRAIL levels were also elevated to a degree similar to that of the total cellular level in BS-181-treated JT/Neo; however, these phenomena were not observed in JT/BCL-2 cells, despite the presence of an approximately 28-fold higher level of total cellular TRAIL in the latter (Figure 2g). Under these conditions, BS-181-induced c-FLIP_L_ down-regulation, caspase-8 activation, BID cleavage, caspase-9/caspase-3 activation, and PARP cleavage—but not DR4/DR5 upregulation—were abrogated in JT/BCL-2 cells. As shown in Figure 3a–e, cell surface staining data also demonstrated a dose-dependent enhancement of TRAIL and DR4/DR5 surface levels in BS-181-treated JT/Neo cells; however, the elevation patterns were not or barely detected on JT/BCL-2 cells.

These results suggested that the BS-181-induced intrinsic BAK-dependent mitochondrial apoptosis pathway might be preceded by extrinsic TRAIL/DR4/DR5 upregulation-mediated and cFLIP_L_ down-regulation-mediated apoptosis pathway in JT/Neo cells, which can be targeted by the anti-apoptotic function of overexpressed BCL-2.

### 2.3. BS-181 Preferentially Induces Apoptosis in G_1_–Arrested JT/Neo Cells after Attenuation of CDK7-Mediated Retinoblastoma (Rb) Inactivation and Cell Cycle Progression

To examine whether BS-181-induced apoptosis is correlated with G_1_ arrest consequent to the inhibition of CDK7 activity, the effect of BS-181 on CDK7-directed activating phosphorylation of CDK1 (Thr161) and CDK2 (Thr160) [31] and CDK-directed phosphorylation of Rb [32,33] in JT/Neo and JT/BCL-2 cells was compared. Although BS-181-induced G_1_ arrest was observed only in JT/BCL-2 cells, CDK7-directed activating phosphorylation of CDK1 (Thr161) and CDK2 (Thr160), and CDK1/2- and CDK4/6-preferred Rb phosphorylation at Thr821/826 and Ser795, respectively, were reduced in both JT/Neo and JT/BCL-2 cells (Figure 4a). Simultaneously, a significant decrease in G_1_-cyclin (cyclin D1 and D3), G_1_-CDK (CDK4), and M-cyclin (cyclin B1) levels was detected in JT/Neo cells but not or barely in JT/BCL-2 cells (Figure 4b). These results indicated that BS-181-induced G_1_ arrest in Jurkat T cells, which was detected only upon blockage of apoptosis by BCL-2 overexpression, was exerted by the failure of Rb phosphorylation by CDKs, resulting from BS-181-mediated effective inhibition of CDK7 catalytic activity rather than down-regulation of G_1_-cyclin (cyclin D1 and D3) and G_1_-CDK (CDK4) levels.

To examine whether BS-181-induced G_1_-arrest serves as a casual event underlying BS-181-induced apoptosis in Jurkat T cells, whether forced arrest at G_1_ phase by hydroxyurea (HU), which blocks the cell cycle in late G_1_ phase by inhibiting the ribonucleotide reductase-mediated conversion of ribonucleotides to deoxyribonucleotides required for DNA synthesis [34,35], could augment BS-181-induced apoptotic processes was investigated.

JT/Neo cells treated with 1 mM HU alone exhibited 68.9% of the cells in the G_1_ phase, 17.1% of the cells in the S phase, 6.8% of the cells in the G_2_/M phase, and 6.5% of the cells in the apoptotic sub-G_1_ phase, suggesting that the majority of the cells became arrested at the G_1_/S border (Figure 5a,b). JT/Neo cells treated with 15 μM BS-181 alone caused 31.2% of the cells to be in the apoptotic sub-G_1_ phase, along with 51.4%, 8.6%, and 8.0% of the cells in the G_1_, S, and G_2_/M phases, respectively, whereas cotreatment with 15 μM BS-181 and 1 mM HU yielded an approximately 2.0-fold increase in both apoptotic sub-G_1_ rate and ΔΨ_m_ loss (Figure 5c,d). Western blot analyses also revealed that the presence of HU augmented BS-181-induced apoptotic responses, including DR5 upregulation, c-FLIP_L_ down-regulation, caspase-8 activation, BID cleavage, caspase-3 activation, and PARP degradation (Figure 5e). At the same time, BS-181-induced upregulation of cell-surface DR4, DR5, and TRAIL levels was markedly enhanced by HU (Figure 5f–k).

On the other hand, when BS-181-induced apoptotic cells (stained with FITC–annexin V) were further analyzed for cell cycle state based on DNA content as measured by PI fluorescence intensity, 37.1%, 47.4%, 11.0%, and 3.8% of FITC–annexin V-positive apoptotic cells were in the sub-G_1_, G_1_, S, and G_2_/M phases, respectively, indicating that BS-181-induced apoptotic cells represented mostly those in the sub-G_1_ and G_1_ phases (Figure 6).

These results indicated that BS-181-induced apoptosis in JT/Neo cells was preferentially provoked in G_1_-arrested cells, consequent to the inhibition of CDK-directed Rb phosphorylation required for entry into S phase, and suggested that the G_1_ phase was more susceptible to BS-181-induced TRAIL/DR4/DR5 upregulation and c-FLIP_L_ down-regulation, which caused BCL-2-sensitive mitochondrial apoptotic events, such as ΔΨ_m_ loss and caspase-9/caspase-3 activation, leading to PARP degradation.

### 2.4. BS-181-Induced Apoptosis Occurs Primarily through Extrinsic TRAIL/DR5-Mediated Apoptotic Signaling and Subsequent BAK-Dependent Mitochondrial Apoptosis Induction

To examine whether BS-181-induced activation of the mitochondrial apoptosis pathway is provoked either as downstream event of the extrinsic apoptosis or independent of extrinsic apoptosis, induced cytotoxicity and apoptotic events in BS-181-treated Jurkat clone A3 (wild-type) were compared to those in Jurkat clones I2.1 and I9.2, which are known to fail to activate the extrinsic apoptosis pathway due to FADD and caspase-8 deficiencies, respectively [36].

When A3 cells were treated with 5, 10, 15, and 20 µM BS-181 for 20 h, the cell viabilities declined to levels of 91.7%, 70.4%, 46.0, and 30.7%, respectively. However, the viabilities of I2.1 and I9.2 cells were reduced to a significantly lesser extent compared to that of A3 cells (Figure 7a). The results of flow cytometric analysis showed that the apoptotic sub-G_1_ ratios in A3, I2.1, and I9.2 cells after treatment with 15 μM BS-181 for 20 h were 47.2%, 13.4%, and 13.2%, respectively, with ΔΨ_m_ loss rates of 46.0%, 14.4%, and 16.0% and concomitant BAK activation ratios of 44.3%, 15.8%, and 16.3% (Figure 7b–g). Under the same conditions, BS-181-caused G_1_ cell cycle arrest was not observed in A3 cells, but observed in I2.1 and I9.2 cells where the apoptotic cell death occurred to a significantly lesser extent compared to A3 cells. These results suggested that although BS-181-induced BAK-dependent mitochondrial apoptosis occurred primarily as a downstream event of the extrinsic apoptosis pathway, it could also occur independently of the extrinsic apoptosis pathway when cells were treated with ≥15 μM BS-181.

Western blot analyses showed that BS-181-induced upregulation of DR4/DR5 was prominently detected in A3 cells but not or barely detected in I2.1 and I9.2 cells (Figure 7h). Additionally, caspase-8 activation, BID cleavage, caspase-9/caspase-3 activation, and PARP cleavage were detected at markedly lower levels in I2.1 and I9.2 cells compared to A3 cells. The cell-surface staining data demonstrated that BS-181-induced elevation of surface TRAIL and DR4/DR5 levels was prominently detected in A3 cells; the elevation patterns were barely detected in I2.1 and I9.2 cells, consistent with a published paper showing that the expression levels of TRAIL and DR5 are positively regulated by the extrinsic apoptosis pathway [37] (Appendix A). These results confirmed that BS-181-induced apoptosis was mainly mediated by TRAIL/DR4/DR5 upregulation, which caused the initiation of the extrinsic apoptosis pathway, leading to caspase-8 activation, BAK activation, and mitochondria-dependent activation of the caspase cascade.

Due to the critical role of extrinsic apoptosis in BS-181-mediated cytotoxicity in Jurkat cells, it was highly likely that the exogenous addition of rTRAIL might augment BS-181-induced extrinsic TRAIL/DR-mediated apoptosis and thus reduce the BS-181 concentration required for inducing apoptosis in Jurkat T cells. To test this prediction, the effect of exogenously added rTRAIL on BS-181 cytotoxicity was investigated in A3 cells. As a result, rTRAIL (1 ng/mL) addition provoked a synergistic effect on 10 μM BS-181-mediated cytotoxicity, confirming that BS-181-induced extrinsic apoptosis could be augmented by adding rTRAIL in Jurkat cells (Figure 8a). To elucidate the mechanisms underlying the synergistic effect of combined BS-181 and rTRAIL treatment, the effect of rTRAIL on BS-181-induced extrinsic apoptotic responses in A3 cells was investigated. After A3 cells with 10 μM BS-181, 10 μM BS-181 plus 1 ng/mL rTRAIL, and 1.0 ng/mL rTRAIL for 15 h, the ratios of sub-G_1_ cells were 12.1%, 40.5%, and 18.9%, the ΔΨ_m_ loss rates were 10.7%, 46.7%, and 24.2%, and the BAK activation rates were 11.7%, 38.4%, and 13.9%, respectively (Figure 7b–g). Western blot analysis also revealed that BS-181-induced DR5 upregulation, caspase-8 activation, and BID cleavage were significantly enhanced by the combined treatment with rTRAIL, as were caspase-9/caspase-3 activation and PARP cleavage (Figure 8h). 

As the cell-surface levels of DR4 and DR5 were commonly upregulated after BS-181 treatment, whether the synergic apoptotic effect of the combined BS-181 and rTRAIL treatment was exerted by DR4 and/or DR5 was examined. When the effect of DR4- and DR5-specific blocking antibodies on the cytotoxicity of the combined BS-181 and rTRAL treatment was investigated, the presence of the DR5-specific blocking antibody, but not the DR4-specific blocking antibody, was able to significantly attenuate the cotreatment cytotoxicity toward A3 cells (Figure 8i).

These results further demonstrated that the BS-181 cytotoxicity mainly proceeded by the extrinsic TRAIL/DR5-dependent apoptosis pathway and that the synergistic effect of rTRAIL on BS-181 cytotoxicity in A3 cells resulted from the augmentation of the extrinsic TRAIL/DR5-mediated apoptotic signaling and subsequent BAK-dependent mitochondrial apoptosis induction.

### 2.5. rTRAIL Exhibits No Synergistic Effects on BS-181 Cytotoxicity in Normal Human T Cells

To examine whether BS-181 cytotoxicity discriminates malignant and normal cells and whether selective BS-181 cytotoxicity toward malignant tumor cells can be enhanced by the combined treatment with rTRAIL, the individual effects of BS-181 in either the absence or presence of rTRAIL on malignant cells (Jurkat A3, U937, and HeLa) and normal human T cells (unstimulated and PHA-stimulated peripheral T cells) were investigated by MTT assay. The IC_50_ values of BS-181 against Jurkat A3, Jurkat JT/Neo, Jurkat JT/BCL-2, Jurkat J/Neo, Jurkat J/BCL-XL, U937, and HeLa cells were 14.5, 15.0, 50.8, 14.2, 48.1, 16.4, and 17.3 μM, respectively; however, those against unstimulated and PHA-stimulated normal human T cells were 43.9 and 18.9 μM, respectively (Table 1). These results indicated that although unstimulated normal human T cells were significantly less sensitive to BS-181 cytotoxicity than malignant tumor cells (Jurkat A3, U937, and HeLa), PHA-stimulated normal human T cells in an interleukin-2 (IL-2)-driven rapidly proliferating state exhibited a slightly lower level of sensitivity than those of the malignant cells tested.

When the effect of the exogenous addition of rTRAIL (1–4 ng/mL) on the cytotoxicity of BS-181 (2.5, 5.0, 7.5, 10, and 15 μM) was compared among malignant tumor cells (Jurkat A3, U937, and HeLa), the synergistic effect of rTRAIL on BS-181 cytotoxicity was confirmed in these tumor cells tested, albeit to varying degrees (Figure 9). Moreover, the exogenous addition of rTRAIL at 4 ng/mL concentration was able to lower the IC_50_ values of BS-181 against Jurkat A3, U937, and HeLa cells to 3.4, 11.4, and 13.0 μM, respectively, indicating that the synergistic cytotoxic effect of rTRAIL and BS-181 in combination was the most remarkable in Jurkat T cells. When the cell-surface TRAIL, DR4, and DR5 levels were investigated in U937 and HeLa cells following treatment with 15 and 20 μM BS-181, the BS-181-induced alterations in their levels appeared to be differential in U937 and HeLa cells (Appendix A). Following BS-181 treatment, the cell-surface DR4/DR5 levels were markedly enhanced except for the TRAIL level which remained constant in U937 cells, whereas the cell surface TRAIL and DR4/DR5 levels were enhanced but not remarkably in HeLa cells. These results suggested that the differences in BS-181-induced upregulation of cell-surface TRAIL and DR4/DR5 levels resulted in more remarkable cytotoxicity of combined rTRAIL and BS-181 treatment in Jurkat T cells than U937 and HeLa cells.

The IL-2-dependent proliferation of PHA-stimulated normal human T cells was barely affected by BS-181 cytotoxicity at concentrations up to 10 μM, exhibiting ≈70.4% cell viability at 15 μM, whereas the synergistic effect of rTRAIL (1–4 ng/mL) on BS-181 cytotoxicity was not observed in PHA-stimulated normal human T cells (Figure 10a). Under the same conditions, there was a remarkable synergistic effect observed in malignant Jurkat A3 cells (Figure 10b). To determine the mechanism underlying the resistance of normal human T cells against the cytotoxic effects of BS-181 or the synergistic effect of the combined BS-181 and rTRAIL treatment, the cell-surface levels of TRAIL and DR5 were compared between PHA-stimulated normal human T cells and Jurkat A3 cells treated with BS-181 at concentrations of 10 and 15 μM. As shown in Figure 10c–f, the cell-surface expression levels of TRAIL and DR5, which were significantly enhanced in Jurkat A3 cells after BS-181 treatment, was not or barely affected by BS-181 treatment, suggesting that the failure to up-regulate cell surface TRAIL and/or DR5 levels in BS-181-treated normal human T cells was responsible for the resistance of normal T cells against the synergistic cytotoxic effect exerted by cotreatment of BS-181 and rTRAIL.

These results suggest that the BS-181 and rTRAIL combination may be promising as a chemotherapeutic strategy in T-ALL, because cotreatment with rTRAIL could lower the BS-181 concentration required to attain IC_50_ values against malignant Jurkat T cells to a pharmacologically safe level, which failed to disturb the proliferation of activated normal human T cells.

## 3. Discussion

This is the first report to demonstrate that the antitumor activity of the CDK7 inhibitor BS-181 is exerted by G_1_-cell cycle arrest and TRAIL/DR5 upregulation-mediated initiation of the extrinsic pathway, and subsequent activation of the intrinsic mitochondria-dependent pathway of apoptosis, which was preferentially provoked in G_1_ phase, in human T-ALL Jurkat T cells. Previously, BS-181 was reported to induce G_1_-arrest and apoptosis, resulting from the down-regulation of G_1_ cyclin (cyclin D1) and anti-apoptotic proteins (MCL-1 and XIAP) in breast cancer cells [21] and down-regulation of cyclin D1 and XIAP in gastric cancer cells [22]. In this study, BS-181-induced G_1_-arrest was not observed in Jurkat T cells unless BS-181-induced apoptosis was inhibited either by overexpression of BCL-2 or BCL-XL or by impairments (deficiency of FADD or caspase-8) of the extrinsic apoptotic pathway. After treatment with BS-181, the levels of CDK7-directed activating phosphorylation of CDK1 (Thr161) and CDK2 (Thr160), and the resultant CDK1/2- and CDK4/6-preferred phosphorylation of Rb, were significantly reduced in Jurkat T cells regardless of BCL-2 overexpression. At the same time, marked decreases in G_1_-cyclin (cyclin D1 and D3), G_1_-CDK (CDK4), and M-cyclin (cyclin B1) levels were detected in JT/Neo cells but not or barely in JT/BCL-2 cells. As the Rb function that blocks the cell cycle at the G_1_ checkpoint is compromised to allow G_1_/S transition when hyperphosphorylated by CDKs [38], BS-181-induced G_1_-arrest observed in JT/BCL-2 cells likely occurred as a consequence of the failure of Rb phosphorylation by CDKs. It was also likely that the majority of the JT/Neo cells arrested in the G_1_ phase, due to impaired Rb phosphorylation, might rapidly undergo apoptotic cell death. Previously, the advantage of BCL-2 overexpression was reported in dissecting the correlation between p53-mediated G_1_-arrest and p53-mediated apoptosis in murine M1 myeloid leukemia cells, in which p53-mediated G_1_-arrest was not detected unless the simultaneous induction of p53-mediated apoptosis was delayed by BCL-2 overexpression [39].

To further examine the correlation between BS-181-mediated G_1_-arrest and apoptosis induction, not only was the impact of forced G_1_-arrest by the late G_1_-blocking agent HU [34,35] on BS-181-induced apoptosis investigated, but the cell cycle distribution of BS-181-induced FITC–annexin V-positive apoptotic cells based on the DNA content measurable by PI staining in JT/Neo cells was too. We found that BS-181-induced upregulation of TRAIL, DR4, and DR5 levels, and the down-regulation of c-FLIP, which can render the initiation of the extrinsic apoptotic pathway, were markedly enhanced in the presence of HU. Additionally, BS-181-induced FITC–annexin V-positive apoptotic cells represented mostly those in the sub-G_1_ and G_1_ phases. These results demonstrated that BS-181-induced apoptotic cell death was preferentially provoked in G_1_-phase cells and supported our prediction that the majority of the JT/Neo cells being arrested in the G_1_ phase after BS-181 treatment were eliminated by apoptotic cell death.

The contribution of the extrinsic apoptotic mechanisms to BS-181-induced apoptogenic activity toward Jurkat T cells in this study was first predicted by Western blot analysis and cell-surface immunofluorescence staining data showing a significant increase in the levels of DR5/TRAIL in BS-181-treated JT/Neo cells. This prediction was further supported by findings that whereas wild-type Jurkat A3 clone was sensitive to the cytotoxicity of BS-181 at concentrations of 10–20 μM and exhibited 70.4–30.7% cell viabilities, Jurkat FADD-deficient I2.1 and caspase-8-deficient I9.2 clones, which cannot transduce the extrinsic apoptotic signals provoked by the interaction of TRAIL with DR5 [36,40], were largely resistant to cytotoxicity. In addition, after exposure to 15 μM BS-181, both I2.1 and I9.2 cells appeared to induce intrinsic apoptosis via the BAK activation-mediated mitochondrial apoptotic mechanism; however, the proportions of apoptotic sub-G_1_ cells, BAK activation, and ΔΨ_m_ loss reached only approximately 28–34% of the proportions detected in A3 cells. Moreover, the exogenous addition of rTRAIL synergized cytotoxicity exerted by 10 μM BS-181 in A3 cells by markedly potentiating the extrinsic DR5-mediated and subsequent mitochondrial damage-mediated apoptotic responses. It is noteworthy that BS-181 cytotoxicity and the synergic effect of the exogenous addition of TRAIL on BS-181-mediated cytotoxicity were almost completely abrogated in the presence of the DR5-specific blocking antibody but not by the DR4-specific blocking antibody. These results confirmed that extrinsic apoptotic mechanisms provoked by TRAIL/DR5 upregulation and proceeded by the subsequent activation of the mitochondria-dependent pathway played a critical role in the induction of apoptotic cell death in Jurkat T cells treated with BS-181 alone or co-treated with BS-181/TRAIL.

It is also noteworthy that although Western blotting data showed that the total cellular TRAIL levels were 28-fold higher in JT/BCL-2 than JT/Neo cells, no increase in the total cellular, cell surface, or secreted soluble TRAIL levels was observed in JT/BCL-2 cells after BS-181 treatment. Simultaneously, whereas the total cellular levels of DR4/DR5 and BS-181-mediated enhancement were higher in JT/BCL-2 cells than in JT/Neo cells, BS-181-induced elevation of cell-surface DR4/DR5 levels was observed only in JT/Neo cells but not in JT/BCL-2 cells. This suggested that BS-181-induced TRAIL upregulation, and the extracellular secretion of TRAIL and DR4/DR5 for membrane-bound forms, might comprise the primary targets of the anti-apoptotic function of overexpressed BCL-2. However, the underlying mechanism remains to be elucidated.

Given the essential roles of CDK7 in regulating the cell cycle and transcription [18,19,20], the outcomes of its specific inhibition in tumor cells by BS-181, including cell cycle arrest and apoptotic cell death, were also expected in rapidly proliferating normal cells. As expected, unstimulated normal human T cells appeared to be significantly less sensitive to BS-181 cytotoxicity than malignant tumor cells (Jurkat A3, U937, and HeLa), whereas the sensitivity of IL-2-dependent proliferation of PHA-stimulated normal T cells was somewhat close to those of the malignant cells when treated with BS-181 at concentrations above 15 μM. The failure of BS-181 cytotoxicity to distinguish between rapidly proliferating cells of normal tissues and tumor cells was likely to cause normal cell damage leading to side effects when applied to chemotherapy. In this regard, whether cotreatment of TRAIL with BS-181 can reduce the concentration of BS-181 enough to induce cytotoxicity toward malignant tumor cells was examined. Consistent with previous studies showing that TRAIL induces apoptosis selectively in malignant tumor cells but not in normal cells [41,42], the synergistic effect of rTRAIL (1‒4 ng/mL) on BS-181 cytotoxicity was not detected in either unstimulated or PHA-stimulated normal T cells, but was apparently detected in all tested malignant cells with the highest efficacy in Jurkat cells. Notably, the exogenous addition of 4 ng/mL rTRAIL resulted in lowering the IC_50_ values of BS-181 against Jurkat A3 to 3.4 μM, in which IL-2-dependent rapid proliferation of PHA-activated normal human T cells was not impaired. These results demonstrate that combined treatment with rTRAIL may serve as a promising strategy to enhance the antitumor efficacy of the CDK7 inhibitor BS-181 and minimize its toxic side effects.

Furthermore, it has recently been reported that four CDK7 inhibitors (ICEC0942, SY-1365, SY-5609, and LY3405105), which were developed to improve the poor bioavailability and insufficient permeability of BS-181, have progressed to phase I/II clinical trials for evaluating their antitumor efficacy [43]. Although only ICEC0942 among these four CDK inhibitors is a BS-181 analog that has been developed to retain the CDK7 selectivity of BS-181 [44], and is thus highly likely to induce extrinsic apoptosis through TRAIL/DR5 upregulation in tumor cells, the contributions of these CDK7 inhibitors to the extrinsic TRAIL/DR5-dependent apoptosis induction in tumor cells and their synergistic cytotoxic effect exerted by cotreatment with rTRAIL remain to be elucidated.

## 4. Materials and Methods

### 4.1. Reagents and Antibodies

BS-181 HCl was purchased from Selleckchem (Houston, TX, USA), and rTRAIL was obtained from PeproTech (London, UK). A human TRAIL enzyme-linked immunosorbent assay (ELISA) kit was purchased from Koma Biotech Inc. (Seoul, Korea). DiOC_6_, 3-(4,5-dimethythiazol-2-yl)-2,5-diphenyl-tetrazolium bromide (MTT), phytohemagglutinin (PHA), and recombinant human interleukin-2 (rIL-2) were purchased from Sigma-Aldrich (St. Louis, MO, USA). The ECL Western blot kit was purchased from Amersham (Arlington Heights, IL, USA). The annexin V–FITC apoptosis kit was obtained from Clontech (Takara Bio, Shiga, Japan). HU and the active conformation-specific anti-BAK (Ab-1) were obtained from Calbiochem (San Diego, CA, USA). The anti-caspase-3 antibody was purchased from BD Transduction Laboratories (Franklin Lakes, NJ, USA), and anti-DR4 antibody was obtained from Abcam (Cambridge, UK). The anti-BID, anti-caspase-8, anti-caspase-9, anti-p-CDK1 (Tyr15), anti-p-CDK1 (Thr161), anti-p-CDK2 (Thr160), anti-DR5, anti-p-retinoblastoma (Rb) (Ser795), anti-p-Rb (Thr821/826), and anti-TRAIL antibodies were purchased from Cell Signaling Technology (Beverly, MA, USA). The BV421-conjugated anti-DR4 and Alexa Fluor 647-labeled anti-DR5 antibodies for cell surface staining were purchased from BD Biosciences (Chicago, IL, USA). The DR4-specific blocking antibody was obtained from Cell Sciences (Newburyport, MA, USA) and DR5-specific blocking antibody was purchased from R&D Systems (Minneapolis, MN, USA). The other antibodies were purchased from Santa Cruz Biotechnology (Santa Cruz, CA, USA).

### 4.2. Cell Culture

Human acute leukemia Jurkat T cell clones A3, I2.1, and I9.2, and human monoblastic leukemia U937 and human cervical carcinoma HeLa cell lines were purchased from the American Type Culture Collection (Manassas, VA, USA). The leukemia cells and human peripheral T cells were maintained in RPMI 1640 complete medium (Hyclone, Gaithersburg, MD, USA) containing 10% fetal bovine serum (FBS), 20 mM HEPES (pH 7.2), 50 μM 2-mercaptoethanol, and 100 μg/mL gentamicin. HeLa cells were maintained in DMEM (Hyclone, Gaithersburg, MD, USA) supplemented with 10% FBS, 20 mM HEPES (pH 7.2), and 100 μg/mL gentamycin. Jurkat T cell clones stably transfected with an empty vector (JT/Neo) or the *BCL-2* expression vector (JT/BCL-2), an empty vector (J/Neo), or the *BCL-XL* expression vector (J/BCL-XL) were kindly provided by Dr. Dennis Taub (Gerontology Research Center, NIA/NIH, Baltimore, MD, USA). JT/Neo cells, JT/BCL-2 cells, J/Neo cells, and J/BCL-XL cells were maintained in RPMI 1640 complete medium plus 200 μg/mL G418 (A.G. Scientific Inc., San Diego, CA) as described elsewhere [45].

### 4.3. Normal Human Peripheral Blood T Lymphocytes

To prepare human peripheral blood mononuclear cells (PBMCs), heparinized blood obtained from healthy laboratory personnel by venipuncture was centrifuged at 800× *g* for 20 min over Ficoll (Sigma-Aldrich), as previously described [46]. Peripheral T cells were isolated from PBMCs by using a human T cell enrichment column kit (R&D Systems).

### 4.4. Cytotoxicity Assay

BS-181 cytotoxicity was assessed by the MTT assay, as previously described [46].

### 4.5. Flow Cytometric Analysis

The cell cycle state of Jurkat T cells treated with BS-181 was analyzed by the Attune Nxt flow cytometer (ThermoFisher Scientific, Waltham, MA, USA) as described elsewhere [47]. The extent of apoptosis and necrosis was detected using an annexin V-FITC apoptosis kit as described previously [47]. The changes in ΔΨ_m_ after BS-181 treatment were measured after staining with DiOC_6_ [48,49]. BAK activation in Jurkat T cells after BS-181 treatment was measured using an active conformation-specific anti-BAK (Ab-1) as previously described [50]. To detect the cell-surface levels of TRAIL, DR4, and DR5, cells washed with 1× phosphate-buffered saline (PBS) containing 2% FBS were treated with the anti-TRAIL, BV421-conjugated anti-DR4, or Alexa Fluor 647-labeled anti-DR5 antibody for 45 min on ice. The cells treated with anti-TRAIL antibody were rinsed and treated with Alexa Fluor 488-labeled anti-rabbit IgG antibody for an additional 30 min on ice. After fixation in 1% paraformaldehyde on ice for 10 min, cells were washed, and TRAIL, DR4, or DR5-positive cells were measured by flow cytometry.

### 4.6. Preparation of Cell Lysates and Western Blot Analysis

After cell lysate preparation, an equivalent amount of protein lysate (20–25 μg) was electrophoresed on a 4–12% SDS gradient polyacrylamide gel with 3-(N-morpholino) propane sulfonic acid buffer and then electrotransferred to a nylon membrane (Millipore Corporation, Bedford, MA, USA), as described elsewhere [47]. Protein detection was performed with an ECL Western blotting kit, according to the manufacturer’s instructions. Densitometry was performed using ImageQuant TL software (Amersham, Arlington Heights, IL, USA). The arbitrary densitometric units for each protein of interest were normalized to the densitometric units for glyceraldehyde 3-phosphate dehydrogenase (GAPDH).

### 4.7. ELISA for the Detection of Extracellular Soluble TRAIL

The levels of soluble TRAIL in the culture supernatants of Jurkat T cells exposed to BS-181 were quantitatively assayed using a human TRAIL ELISA kit. Measurements were performed according to the manufacturer’s instructions for TRAIL using a 96-well microplate reader at 450 nm. All samples were used in triplicate, and optical density measurements were verified against a standardized curve. The results were averaged and expressed in pg/mL.

### 4.8. Statistical Analysis

Unless otherwise indicated, each result is representative of at least three separate experiments. The values are expressed as the means ± standard deviations (SDs) of the experiments. Statistical significance was calculated using the Student’s t-test. Differences were considered significant at *p* < 0.05.

### 4.9. Study Approval

All procedures for isolation of peripheral T cells from human PBMCs using a human T cell enrichment column kit were approved by the Ethics Committee of Kyungpook National University, Daegu, Korea (KNU2019-0058). Informed written consent was obtained from the participants. All procedures were conducted according to the tenets of the Declaration of Helsinki.

## 5. Conclusions

A validated CDK7 inhibitor BS-181 at a pharmacological dose was found to exert antitumor activity primarily through cell cycle arrest in the G_1_ phase and induction of the extrinsic TRAIL/DR5 upregulation-mediated apoptosis in human T-ALL Jurkat cells (clone A3) with an IC_50_ value of 14.5 μM. The combined treatment with rTRAIL exerted synergistic effects on BS-181 cytotoxicity toward malignant cells (Jurkat A3, U937, and HeLa) but not against normal human peripheral T cells. The combined rTRAIL (1−4 ng/mL) and BS-181 treatment could lower the IC_50_ value of BS-181 against Jurkat A3 cells by augmenting the induction of extrinsic apoptosis to a concentration (3.4 μM) that could spare rapidly proliferating activated normal human T cells, suggesting that the BS-181 and rTRAIL combination is applicable to the clinical treatment of human T-ALL. 

## Figures and Tables

**Figure 1 cancers-12-03845-f001:**
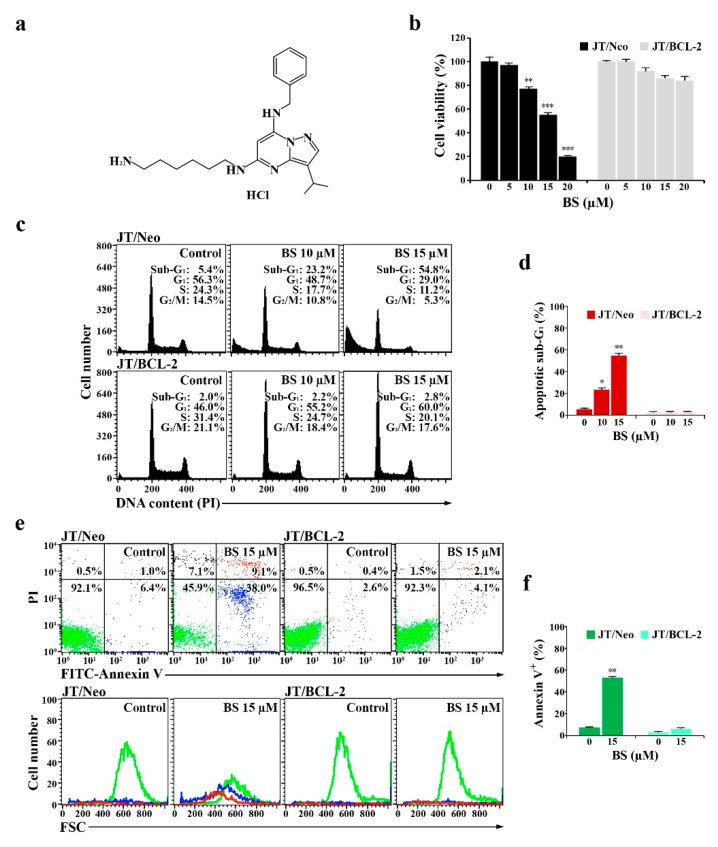
Chemical structure of BS-181 and its effects on cell viability, cell cycle distribution, and apoptotic cell death in Jurkat T cells transfected with the empty vector (JT/Neo) and Jurkat T cells transfected with the *BCL-2* expression vector (JT/BCL-2). (**a**) The validated CDK7 inhibitor BS-181 is a pyrazolo [1,5-α] pyrimidine-derived compound. (**b**) Cell viability was determined by incubating each cell type (5 × 10^4^ cells/well) with the indicated concentrations of BS-181 in a 96-well plate for 20 h and an additional 4 h with MTT solution. Mean ± SD (*n* = 3 with three replicates per independent experiment). ** *p* < 0.01, *** *p* < 0.005, compared with the control. (**c**–**f**) Equivalent cultures were prepared, and cells were collected to analyze cell cycle distribution and apoptosis/necrosis by flow cytometric analyses of PI staining and FITC–annexin V/PI double staining. The FSC properties of individual unstained live cells (green), early apoptotic cells (blue), and late apoptotic cells (red) were measured to analyze the changes in cell size during induced apoptosis. BS: BS-181. Mean ± SD of triplicate experiments. * *p* < 0.05, ** *p* < 0.01, compared with the control.

**Figure 2 cancers-12-03845-f002:**
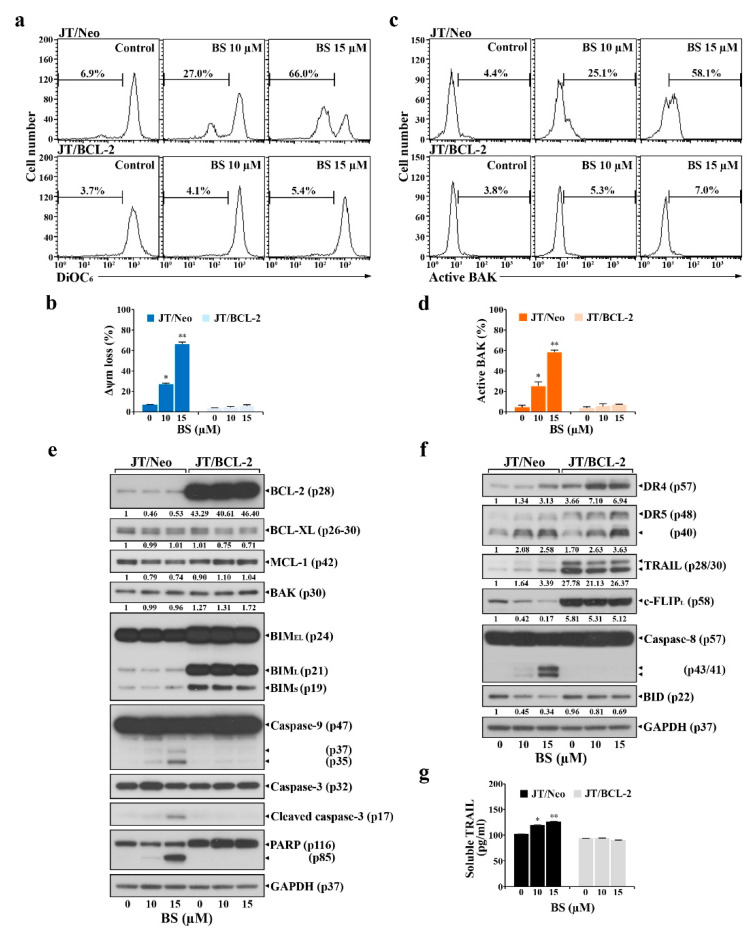
BS-181-induced ΔΨ_m_ loss, BAK activation, and alterations of intrinsic and extrinsic apoptosis-regulating factors in JT/Neo and JT/BCL-2 cells. (**a**–**d**) Each cell type (5 × 10^5^ cells/mL) was treated with 10 to15 μM BS-181 for 24 h and subjected to flow cytometric analysis of ΔΨ_m_ loss and BAK activation, as described in Materials and Methods. Mean ± SD of triplicate experiments. * *p* < 0.05, ** *p* < 0.01, compared with the control. (**e**,**f**) Total cell lysates from equivalent cultures were prepared for Western blot analysis of individual apoptosis-regulating factors (intrinsic: BCL-2, BCL-XL, MCL-1, BAK, BIM isoforms, caspase-9, and caspase-3; extrinsic: DR5, TRAIL, caspase-8, and BID) and GAPDH as described in Materials and Methods. A representative result is shown; two additional experiments yielded similar results. (**g**) After an equivalent culture was prepared, the culture supernatants were subjected to ELISA for detection of secreted TRAIL, as described in Materials and Methods. The results were averaged and expressed in pg/mL. BS: BS-181. Mean ± SD of triplicate experiments. * *p* < 0.05, ** *p* < 0.01, compared with the control.

**Figure 3 cancers-12-03845-f003:**
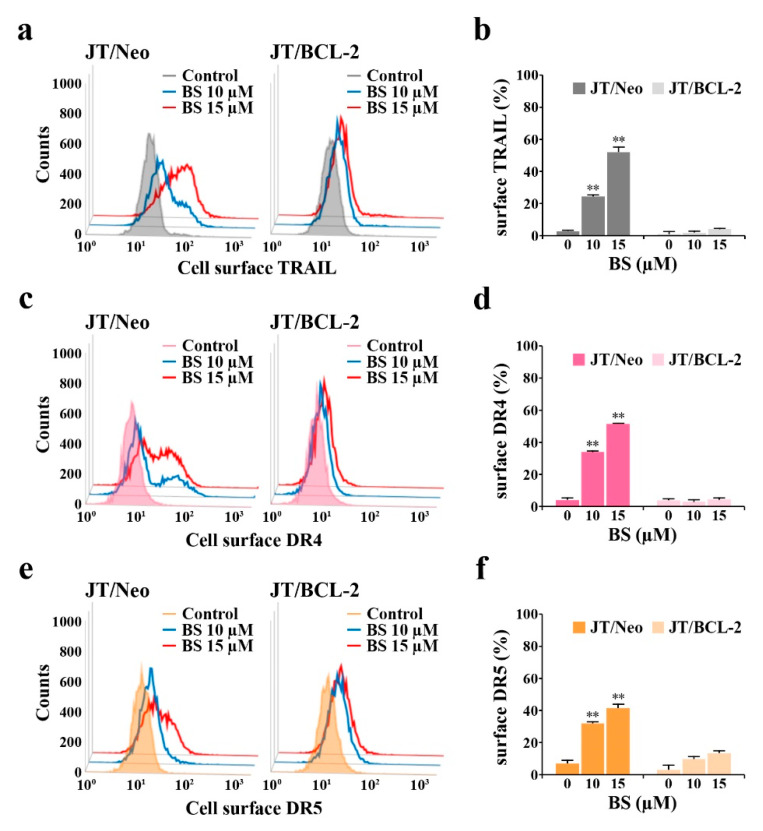
Promotive effect of BS-181 on cell surface levels of TRAIL, DR4, and DR5 in JT/Neo and JT/BCL2 cells. (**a**–**f**) After JT/Neo and JT/BCL-2 cells were exposed to 10 or 15 μM BS-181 for 24 h, cells were harvested for treatment with the anti-TRAIL, BV421-conjugated anti-DR4, or Alexa Fluor 647-labeled anti-DR5 antibody for 45 min on ice. The cells treated with anti-TRAIL antibody were rinsed and treated with Alexa Fluor 488-labeled anti-rabbit IgG antibody for an additional 30 min on ice. After fixation in 1% paraformaldehyde, TRAIL, DR4, or DR5-positive cells were measured by the Attune Nxt flow cytometer, as described in Materials and Methods. Representative results are shown; two additional experiments yielded similar results. BS: BS-181. Mean ± SD of triplicate experiments. ** *p* < 0.01, compared with the control.

**Figure 4 cancers-12-03845-f004:**
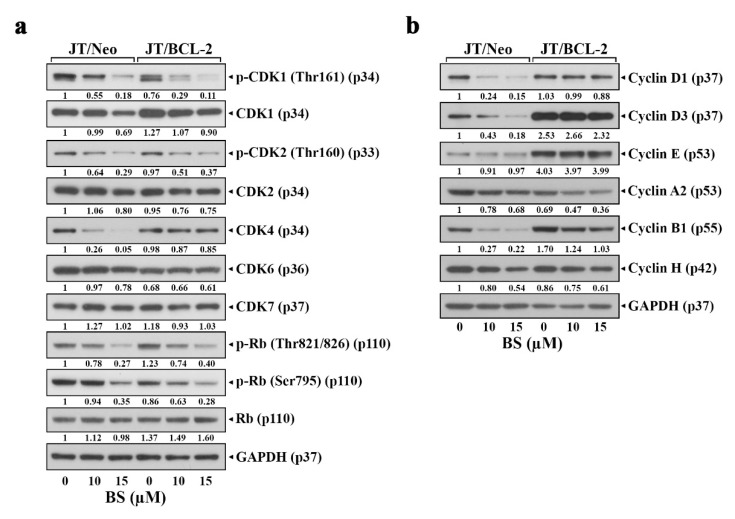
Inhibitory effect of BS-181 on CDK-mediated phosphorylation of Rb in JT/Neo and JT/BCL-2 cells. (**a**,**b**) After each cell type (5 × 10^5^ cells/mL) was treated with 10 to15 μM BS-181 for 24 h, cells were subjected to Western blot analysis of cell cycle-regulating factors including p-CDK1 (Thr161), CDK1, p-CDK2 (Thr160), CDK2, CDK4, CDK7, p-Rb (Thr821/826), p-Rb (Ser795), Rb, cyclin D1, cyclin D3, cyclin E, cyclin A2, cyclin B1, and cyclin H, and GAPDH in BS-181-treated JT/Neo and JT/BCL-2 cells. BS: BS-181. A representative result is shown; two additional experiments yielded similar results.

**Figure 5 cancers-12-03845-f005:**
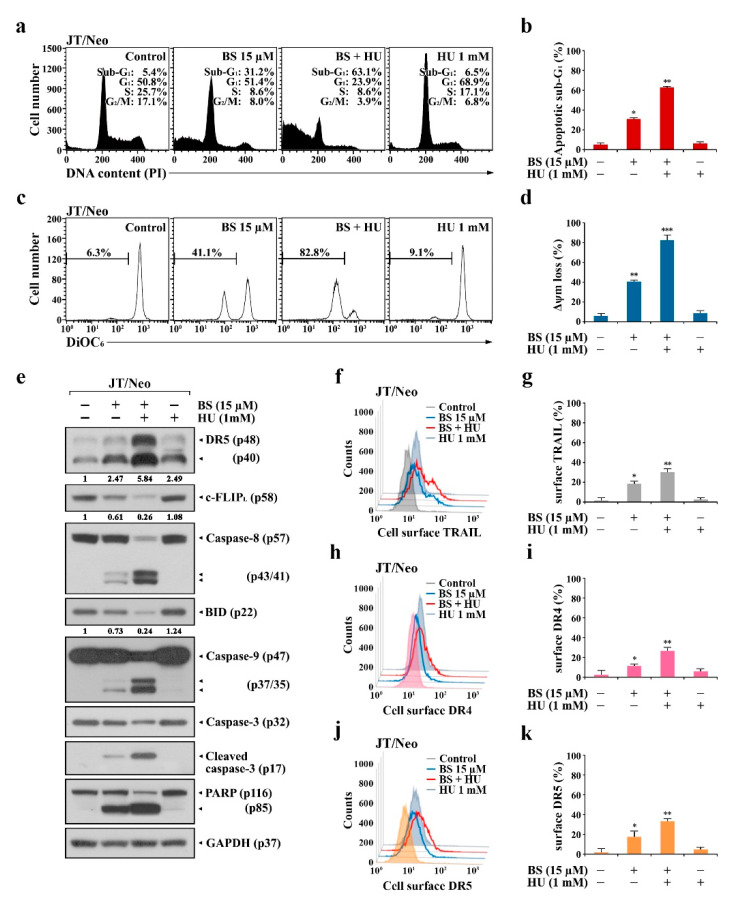
Effect of HU on apoptotic sub-G_1_ accumulation and ΔΨ_m_ loss, FITC–annexin V positive apoptotic cells, and cell surface TRAIL, DR4, and DR5 levels in BS-181-treated JT/Neo cells. (**a**–**d**) JT/Neo cells (5 × 10^5^ cells/mL) were cultured for 12 h in the presence of 15 μM BS-181, 15 μM BS-181 + 1 mM HU, or 1 mM HU. HU was added to the medium 1 h before BS-181 treatment. After the recovery of the cultured cells, PI and DiOC_6_ stainings were performed as described in Materials and Methods. Cell cycle distribution and ΔΨ_m_ loss were examined by flow cytometry analysis. Mean ± SD of triplicate experiments. * *p* < 0.05, ** *p* < 0.01, *** *p* < 0.005, compared with the control. (**e**) Total cell lysates from equivalent cultures were prepared for Western blot analysis of individual apoptosis-regulating factors including DR5, c-FLIP, BID, caspase-9, caspase-3, and PARP, and GAPDH. A representative result is shown; two additional experiments yielded similar results. (**f**–**k**) Individual cells from equivalent cultures were treated with the anti-TRAIL, BV421-conjugated anti-DR4, or Alexa Fluor 647-labeled anti-DR5 antibody for 45 min on ice. The cells treated with anti-TRAIL antibody were rinsed and treated with Alexa Fluor 488-labeled anti-rabbit IgG for an additional 30 min on ice. After fixation in 1% paraformaldehyde, TRAIL, DR4, or DR5-positive cells were measured by the flow cytometry. BS: BS-181. Representative results are shown; two additional experiments yielded similar results. Mean ± SD of triplicate experiments. * *p* < 0.05, ** *p* < 0.01, compared to the control.

**Figure 6 cancers-12-03845-f006:**
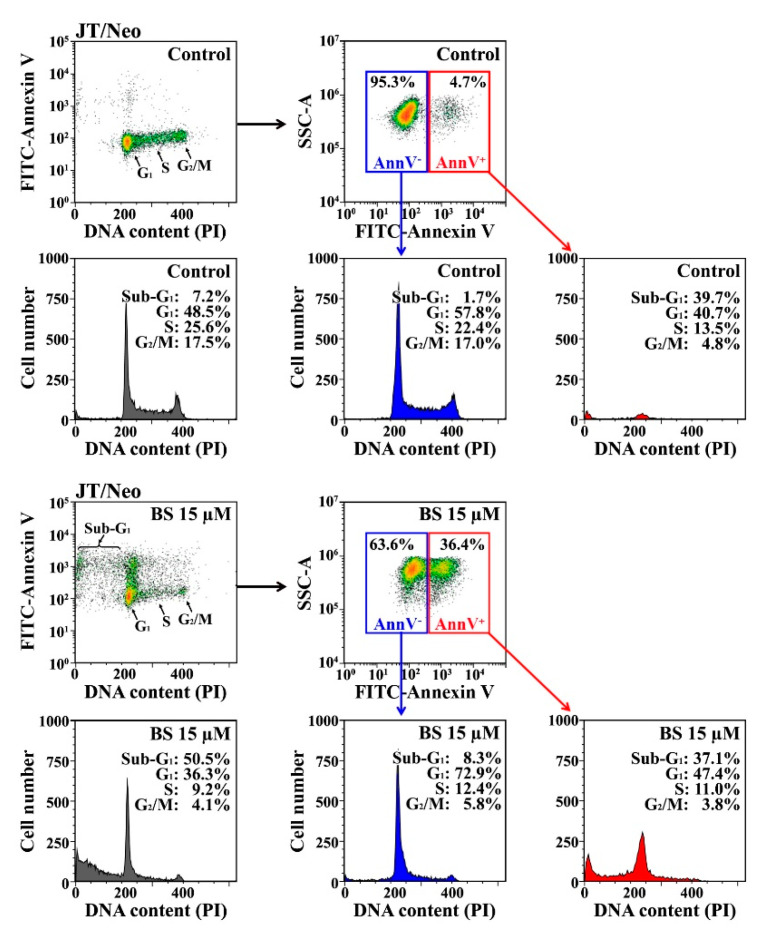
Cell cycle distribution of BS-181-induced FITC–annexin V-positive apoptotic cells in JT/Neo cells. To investigate whether the apoptosis induced by BS-181 treatment preferentially occurs in G_1_-arrested cells, BS-181-treated JT/Neo cells for 24 h were stained with FITC–annexin V, fixed with 1% paraformaldehyde, and then treated with 0.025% digitonin and 50 μg/mL RNase, before staining with PI (25 μg/mL). The cell cycle distribution patterns of FITC–annexin V-positive apoptotic cells were analyzed based on PI fluorescence intensity by flow cytometry. BS: BS-181. A representative result is presented, and two additional experiments yielded similar results.

**Figure 7 cancers-12-03845-f007:**
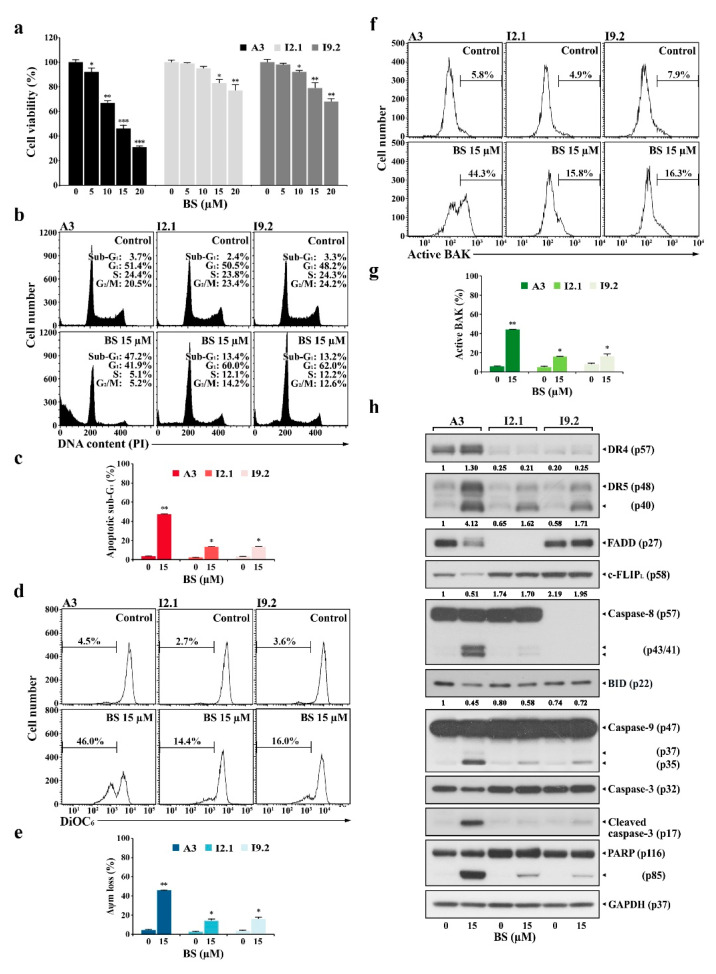
Differential effects of BS-181 on cell viability, cell cycle distribution, ΔΨ_m_ loss, BAK activation, and apoptosis-regulating protein levels in FADD- and caspase-8-positive wild-type, FADD-deficient, and caspase-8-deficient Jurkat T cells. (**a**) Wild-type (clone A3), FADD-deficient (clone I2.1), and caspase-8-deficient (clone I9.2) Jurkat T cells were incubated at a density 5 × 10^4^ cells/well with indicated concentrations of BS-181 in a 96-well plate for 20 h and an additional 4 h with MTT solution to assess cell viability. Mean ± SD (*n* = 3 with three replicates per independent experiment). * *p* < 0.05, ** *p* < 0.01, *** *p* < 0.005, compared with the control. (**b**–**g**) Equivalent cultures were prepared, and cells were collected to analyze cell cycle distribution, ΔΨ_m_ loss, and BAK activation by flow cytometric analyses. Mean ± SD of triplicate experiments. * *p* < 0.05, ** *p* < 0.01, compared with the control. (**h**) Western blot analysis of DR4, DR5, FADD, BID, caspase-8, caspase-9, caspase-3, PARP, and GAPDH. BS: BS-181. A representative result is presented, and two additional experiments yielded similar results.

**Figure 8 cancers-12-03845-f008:**
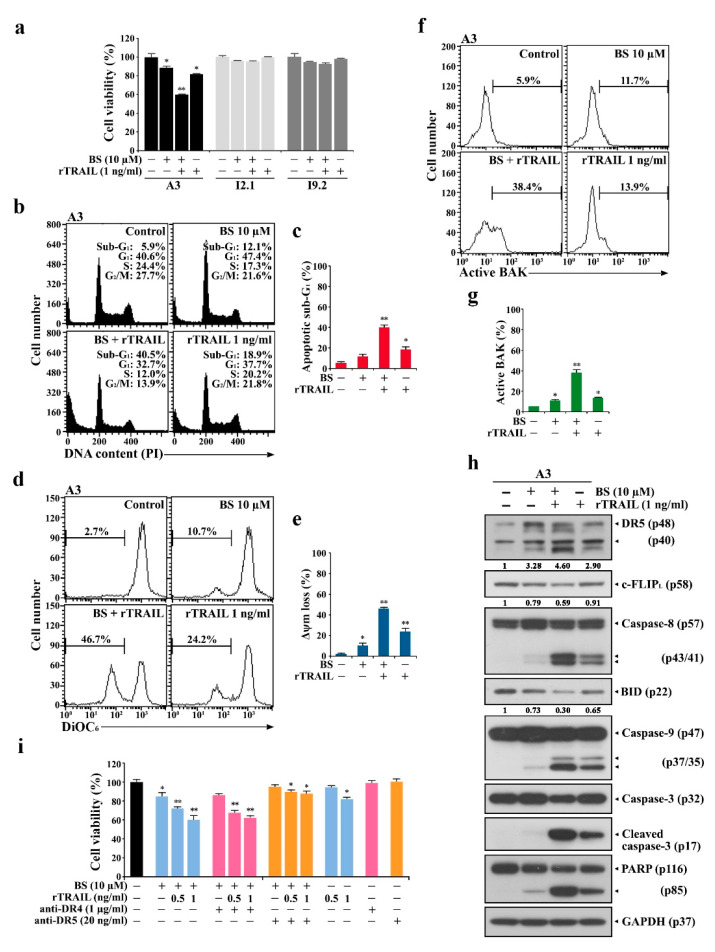
Synergistic effects of cotreatment of BS-181 and rTRAIL on cell viability, apoptotic sub-G_1_ cell accumulation, ΔΨ_m_ loss, and apoptosis induction, and suppressive effect of DR5-specific blocking antibodies on the cotreatment cytotoxicity in Jurkat T cells. (**a**) Cell viability was determined by incubating wild-type A3, FADD-deficient I2.1, and caspase-8-deficient I9.2 cells (5 × 10^4^ cells/well) with 10 μM BS-181, 1 ng/mL rTRAIL, or 10 μM BS-181 plus 1 ng/mL rTRAIL in a 96-well plate for 12 h and an additional 4 h was incubated with MTT solution to assess cell viability. Mean ± SD (*n* = 3 with three replicates per independent experiment). * *p* < 0.05, ** *p* < 0.01, compared with the control. (**b–g**) After A3 cells (5 × 10^5^ cells/mL) was treated with 10 μM BS-181, 1 ng/mL rTRAIL, or 10 μM BS-181 + 1 ng/mL rTRAIL for 16 h, cells were subjected to flow cytometric analysis of cell cycle distribution, ΔΨ_m_ loss, and BAK activation. Mean ± SD of triplicate experiments. * *p* < 0.05, ** *p* < 0.01, compared with the control. (**h**) Equivalent cultures were prepared and subjected to Western blot analysis of DR5, c-FLIP_L_, caspase-8, BID, caspase-9, caspase-3, PARP, and GAPDH. A representative result is presented; two additional experiments yielded similar results. (**i**) A3 cells (5 × 10^4^ cells/well) were incubated with 10 μM BS-181, 10 μM BS-181 + 0.5–1 ng/mL rTRAIL, 10 μM BS-181 + 0.5–1 ng/mL rTRAIL + 1 μg/mL DR4-blocking antibody, or 10 μM BS-181 + 0.5–1 ng/mL rTRAIL + 20 ng/mL DR5-blocking antibody in a 96-well plate for 12 h and with MTT solution for an additional 4 h to evaluate cell viability. BS: BS-181. Mean ± SD (*n* = 3 with three replicates per independent experiment). * *p* < 0.05, ** *p* < 0.01, compared with the control.

**Figure 9 cancers-12-03845-f009:**
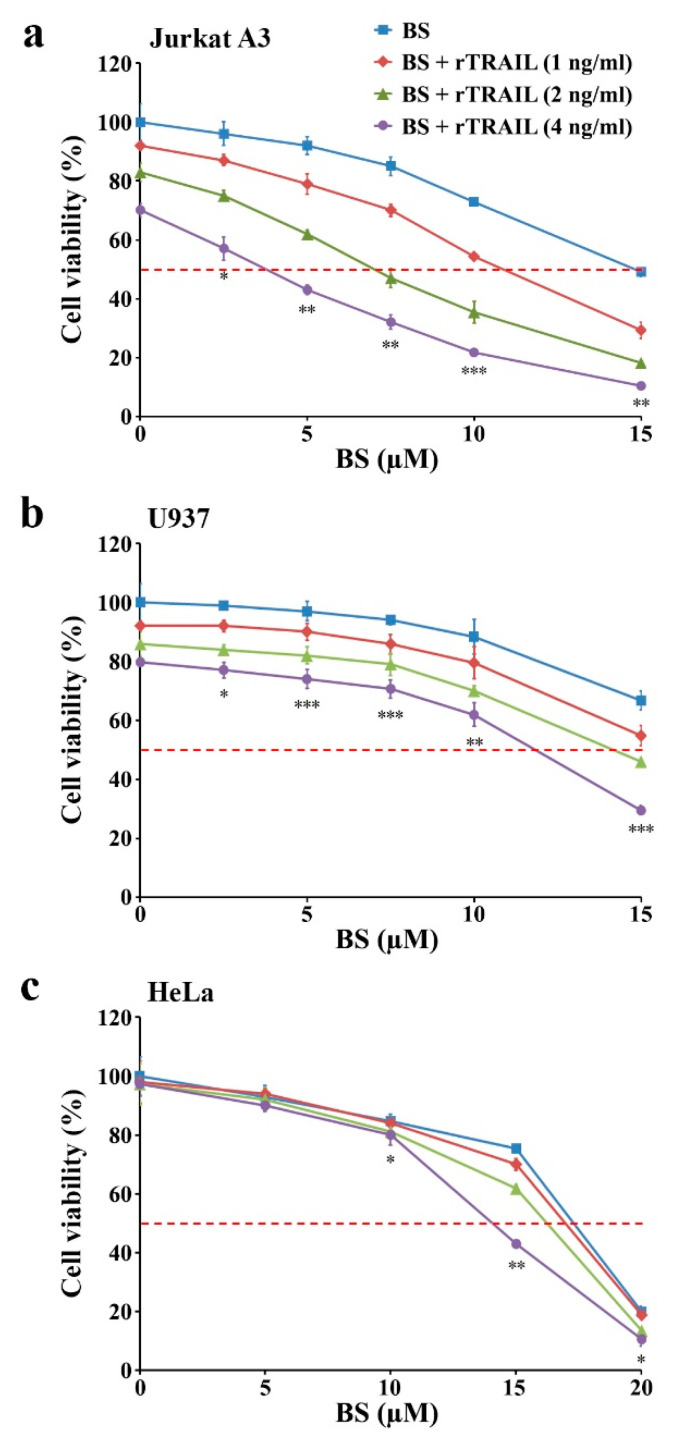
Effects of combination treatments with BS-181 and rTRAIL on Jurkat A3, U937, and HeLa. (**a**–**c**) Jurkat A3 cells (5 × 10^4^ cells/well), U937 cells (5 × 10^4^ cells/well), and HeLa cells (5 × 10^3^/well) were incubated with indicated concentrations of BS-181 and rTRAIL in a 96-well plate for 20 h and an additional 4 h with MTT solution to assess cell viability. BS: BS-181. Mean ± SD (*n* = 3 with three replicates per independent experiment). * *p* < 0.05, ** *p* < 0.01, *** *p* < 0.005, compared with the control.

**Figure 10 cancers-12-03845-f010:**
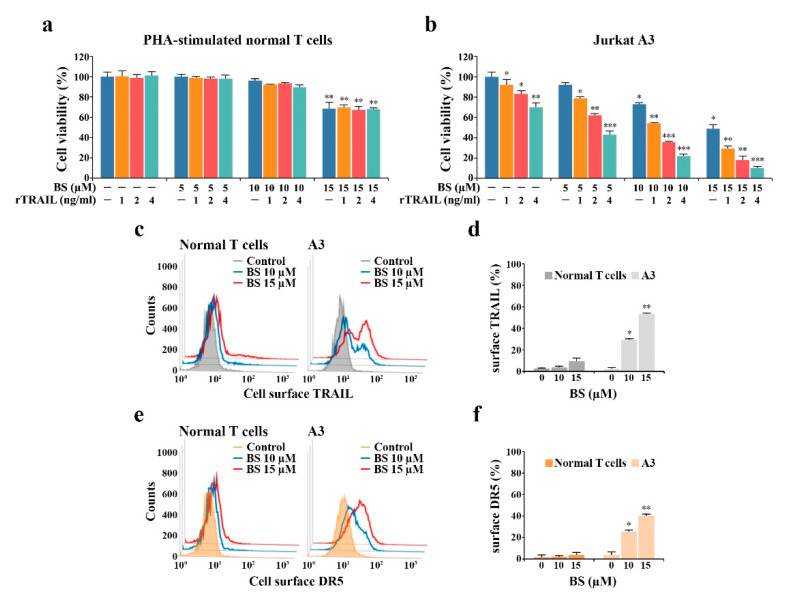
Effect of the combined BS-181 and rTRAIL treatment on normal peripheral T lymphocytes. (**a**) To measure the effect of BS-181 and rTRAIL in combination on IL-2-dependent proliferation of activated normal human T cells, human peripheral T cells after stimulation with 1.0 μg/mL PHA for 60 h were incubated at a cell density of 5 × 10^4^ cells/well with indicated concentrations of BS-181 and rTRAIL in the presence of 25 U/mL of rIL-2 in a 96-well plate for 20 h and with MTT solution for an additional 4 h. Mean ± SD (*n* = 3 with three replicates per independent experiment). ** *p* < 0.01, compared with the control. (**b**) The effect of BS-181 and rTRAIL in combination on Jurkat A3 cells was measured by incubating cells (5 × 10^4^ cells/well) with indicated concentrations of BS-181 and rTRAIL in a 96-well plate for 20 h and an additional 4 h was incubated with MTT solution; Mean ± SD (*n* = 3 with three replicates per independent experiment). * *p* < 0.05, ** *p* < 0.01, *** *p* < 0.005, compared with the control. (**c**–**f**) Individual cells from equivalent cultures were treated with anti-TRAIL or Alexa Fluor 647-labeled anti-DR5 antibody for 45 min on ice. The cells treated with anti-TRAIL antibody were rinsed and treated with Alexa Fluor 488-labeled anti-rabbit IgG antibody for an additional 30 min on ice. After fixation in 1% paraformaldehyde, TRAIL or DR5-positive cells were measured by the flow cytometry. BS: BS-181. Representative results are shown; two additional experiments yielded similar results. Mean ± SD of triplicate experiments. * *p* < 0.05, ** *p* < 0.01, compared to the control.

**Table 1 cancers-12-03845-t001:** Inhibitory effect of BS-181 on unstimulated human peripheral T cells, IL-2-dependent proliferation of PHA-stimulated peripheral T cells, and proliferation of tumor cells.

Tumor and Normal Cells	IC_50_ (μM) *
Unstimulated human peripheral T cells	43.9
PHA-stimulated human peripheral T cells **	18.9
Human acute T cell leukemia Jurkat (JT/Neo) cells	15.0
Human acute T cell leukemia Jurkat (JT/BCL-2) cells	50.8
Human acute T cell leukemia Jurkat (J/Neo) cells	14.2
Human acute T cell leukemia Jurkat (J/BCL-XL) cells	48.1
Human acute T cell leukemia Jurkat A3 cells	14.5
Human monoblastoid U937 cells	16.4
Human cervical carcinoma HeLa cells	17.3

* The IC_50_ value indicates a concentration of BS-181, which caused a 50% reduction in cell viability based on the MTT assay. The cells (Jurkat and U937, 5 × 10^4^ cells/well; HeLa, 5 × 10^3^ cells/well) were cultured with different concentrations of BS-181 for 24 h, and for the final 4 h they were incubated with MTT solution to assess cell viability. ** To induce IL-2-dependent proliferation of stimulated T cells, human peripheral T cells after stimulation with PHA (1.0 μg/mL) for 60 h were harvested and then incubated at a density of 1 × 10^5^ cells/well with different concentrations of BS-181 in the presence of 25 U/mL of recombinant human IL-2 (Sigma-Aldrich) in 96-well plates.

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
