# Peer review of "G1 Cell Cycle Arrest and Extrinsic Apoptotic Mechanisms Underlying the Anti-Leukemic Activity of CDK7 Inhibitor BS-181"

_cancers, 2020, doi:10.3390/cancers12123845_

Round 1

Reviewer 1 Report

The author presents the G1-Cell Cycle Arrest and Extrinsic Apoptotic Mechanisms Underlying Anti-Leukemic Activity of CDK7 Inhibitor BS-181. The work is potentially interesting and the manuscript is well written with only minor grammatical errors. I accept the manuscript to be published after addressing the following issues.

  • In page 2 and line 59, it must be “a part of”.
  • In page 2 and line 72, it must be “might be applied”.
  • In page 2 and line 79, one of the “either” must be deleted.
  • In page 3 and line 95, there must not be a space in “G1 –cell”.
  • There must be a standart writing style in subsections (upper or lower case).
  • In Table 1, a different font was used compared to general writing pattern of article.
  • References part must be rewritten due to the typing “enter” button by mistake in the middle of the sentence. (e.g. below)
  1. Remacle, J.; Steinbach, D. Expression profiling of ATP-binding cassette transporters in
  2. childhood T-cell acute lymphoblastic leukemia. Mol. Cancer Ther. 2006, 5, 1986-1994.

Author Response

In page 2 and line 59, it must be “a part of”.

[Answer]

“It has been corrected.”

In page 2 and line 72, it must be “might be applied”.

[Answer]

“It has been corrected.”

In page 2 and line 79, one of the “either” must be deleted.

[Answer]

   “One either has been removed.”

In page 3 and line 95, there must not be a space in “G1 –cell”.

[Answer]

   Based on the comment, all of “G1-cell cycle arrest” in this manuscript have been changed to “G1 cell cycle arrest”.

There must be a standart writing style in subsections (upper or lower case).

[Answer]

Based on the comment, the subsection tile has been changed as follows.

2.2. BCL-2 overexpression abrogates extrinsic TRAIL/DR5 upregulation-mediated apoptosis and subsequent mitochondrial damage-mediated apoptosis in BS-181-treated JT/Neo cells”

“2.2. BCL-2 Overexpression Abrogates Extrinsic TRAIL/DR5 Upregulation-Mediated Apoptosis and Subsequent Mitochondrial Damage-Mediated Apoptosis in BS-181-Treated JT/Neo Cells”

In Table 1, a different font was used compared to general writing pattern of article.

[Answer]

The format of Table 1 has been corrected.

References part must be rewritten due to the typing “enter” button by mistake in the middle of the sentence. (e.g. below)

13.Remacle, J.; Steinbach, D. Expression profiling of ATP-binding cassette transporters in

14.childhood T-cell acute lymphoblastic leukemia. Mol. Cancer Ther. 2006, 5, 1986-1994.

[Answer]

Based on the comment, the references have been retyped properly.

Reviewer 2 Report

The manuscript by Park et al describes the mechanisms of apoptosis induced by CDK7 inhibitor BS-181 in T-ALL Jurkat cells. It is demonstrated that inhibition of CDK7 induces activation of extrinsic pathway of apoptosis by enhancing TRAIL and death receptors DR4 and DR5 surface expression.

Some limitations need to be addressed before publication.

  1. BS-181 induced enhancement of surface expression of death receptors and TRAIL is shown in Jurkat (A3) cells, but not in U937 and HeLa cells. Since the latter are less sensitive to TRAIL, it is necessary to elucidate the role of BS-181 in the activation of the extrinsic pathway of apoptosis in U937 and HeLa cells.
  2. The numbering in the cited literature should be corrected.
  3. There is no legend to the supplementary figure 1.
  4. The sentence in lines 207-211 should be corrected, since it is not clear what it is about.
  5. BS-181 is indicated as BS in all figures. The legend should indicate either that BS-181 is BS, or BS to BS-181mught be changed in the figures.
  6. Abstract: Lines 13-14 “The antitumor activity of the CDK7 inhibitor BS-181 to human T-ALL Jurkat cells was 13 determined”. The antitumor activity means experiments on tumor.

Author Response

  1. BS-181 induced enhancement of surface expression of death receptors and TRAIL is shown in Jurkat (A3) cells, but not in U937 and HeLa cells. Since the latter are less sensitive to TRAIL, it is necessary to elucidate the role of BS-181 in the activation of the extrinsic pathway of apoptosis in U937 and HeLa cells.

[Answer]

To meet this comment, we analyzed the cell-surface levels of TRAIL, DR4, and DR5 in BS-181-treated U937 and HeLa cells. The results have been included as Supplementary 3, and mentioned in the Results section (Page 14, Line 406-414) as follows.

Line 406-414:

“When the cell-surface TRAIL, DR4, and DR5 levels were investigated in U937 and HeLa cells following treatment with 15 and 20 μM BS-181, the BS-181-induced alterations in their levels appeared to be differential in U937 and HeLa cells (Supplementary 3). Following BS-181 treatment, the cell-surface DR4/DR5 levels were markedly enhanced except for the TRAIL level which remained constant in U937 cells, whereas the cell surface TRAIL and DR4/DR5 levels were enhanced but not remarkably in HeLa cells. These results suggested that the differences in BS-181-induced upregulation of cell-surface TRAIL and DR4/DR5 levels resulted in more remarkable cytotoxicity of combined rTRAIL and BS-181 treatment in Jurkat T cells than U937 and HeLa cells.”

2.The numbering in the cited literature should be corrected.

[Answer]

Based on the comment, the references have been retyped properly.

3.There is no legend to the supplementary figure 1.

[Answer]

During the revision, the number of Supplementary figures increased to 3. Individual Supplementary figure legends have been included in the revised manuscript.

4.The sentence in lines 207-211 should be corrected, since it is not clear what it is about.

[Answer]

To meet this comment, the sentences were changed as follows.

Line 267-269:

“When A3 cells were treated with 5, 10, 15, and 20 µM BS-181 for 20 h, the cell viabilities declined to levels of 91.7%, 70.4%, 46.0, and 30.7%, respectively. Under the same conditions, however, the viabilities of I2.1 and I9.2 cells were almost completely refractory to BS-181 cytotoxicity at a 5 μM concentration but declined at concentrations of 10 to 20 μM, albeit to a significantly lesser extent compared to that of A3 cells (Figure 7a).”

→ “ When A3 cells were treated with 5, 10, 15, and 20 µM BS-181 for 20 h, the cell viabilities declined to levels of 91.7%, 70.4%, 46.0, and 30.7%, respectively. However, the viabilities of I2.1 and I9.2 cells were reduced to a significantly lesser extent compared to that of A3 cells (Figure 7a).”

5.BS-181 is indicated as BS in all figures. The legend should indicate either that BS-181 is BS, or BS to BS-181 mught be changed in the figures.

[Answer]

To meet this comment, each figure legend, “BS: BS-181.” has been included.

6.Abstract: Lines 13-14 “The antitumor activity of the CDK7 inhibitor BS-181 to human T-ALL Jurkat cells was 13 determined”. The antitumor activity means experiments on tumor.

[Answer]

To meet this comment, the sentence has been changed as follows.

Line 13-14:

“The antitumor activity of the CDK7 inhibitor BS-181 to human T-ALL Jurkat cells was determined.”

→ “In vitro antitumor activity of the CDK7 inhibitor BS-181 against human T-ALL Jurkat cells was determined.”

Reviewer 3 Report

This manuscript by Park et al, reported that CDK7 Inhibitor BS-181 inhibited human T-ALL Jurkat cell growth through induction of G1 cell cycle arrest and extrinsic apoptosis. This paper presented interesting data and may uncover novel mechanisms of action of this small chemical. Overall this paper was read with enthusiasm, particularly the high quality of WB data. However, there are also some concerns that need to be addressed well.

1) Due to instability of chromosomes in cancer cells, working on single clones has clear limits, and cell populations with targeted gene expression is more reliable. If indeed on single clones, at lease three clones each should be observed. In this study, authors should evaluate the cell viability in two more clones each to confirm the bio-effects, and then extensive studies could work on one or two clones.

2) Sub-G1 in flow data (images) is questionable. Due to apoptosis and DNA fragmentation, Sub-G1 occurs for apoptosis, but this peak is somewhat close to G1, rather than Y-axis. Data here looks more like cell debris. Since BS-181 works in nanomolar (Introduction), uM levels were used by the authors, which may induce vast cell death in short time and production of vast cell debris. Consider to lower down the doses and repeat flow assays.

3) It is also weird that to evaluate cell viability in 20 hours. It is usually in 48 or 72 hours with low doses (nanomolar in literature).

4) IC50 in JT/BCL-2 needed.

5) "G1 arrest ONLY in JT/BCL2..." is questionable. G1 cell percentage decreased in JT/Noe cells due to apoptosis. Low doses in JT/Neo cells is necessary to test the G1 arrest.

6) BS-181 induced extrinsic apoptosis through induction of receptors and ligands. It appear a novel mechanism of action. As a CDK7 inhibitor, how does it work? Hypothetical discussion may need.

7) Refs are completely de-formatted.

Author Response

1) Due to instability of chromosomes in cancer cells, working on single clones has clear limits, and cell populations with targeted gene expression is more reliable. If indeed on single clones, at lease three clones each should be observed. In this study, authors should evaluate the cell viability in two more clones each to confirm the bio-effects, and then extensive studies could work on one or two clones.

[Answer]

This is a very good point. As described in the Materials and Method section, Jurkat T cell clones JT/Neo, JT/BCL-2, J/Neo, and J/BCL-XL were kindly provided by Dr. Dennis Taub (Gerontology Research Center, NIA/NIH, Baltimore, MD, USA). My first research paper using these Jurkat T cell clones were published in Biochem. Biophys. Res. Commun. 295 (2002) 283-288. Since then, several research papers using these Jurkat T cell clones have been continuously published as follows.

------------------------------------------------------------------

Biochemical Pharmacol. 66 (2003) 2291-2300.

Carcinogenesis 28 (2007) 1303-1313.

Food Chem. 100 (2007) 99-107.

Biochem. Biophys. Res. Commun. 377 (2008) 280-285.

Toxicol. Appl. Pharmacol. 241 (2009) 210-220.

Biochemical Pharmacol. 82 (2011) 1110-1125.

Process Biochem. 48 (2013) 945-954.

Biochim. Biophys. Acta 1833 (2013) 2220-2232.

Apoptosis 19 (2014) 224-240.

Biochemical Pharmacol. 94 (2015) 257-269.

Oncotarget 9 (2018) 4969-4984.

PLOS One 13 (2018) e0204585.

Oxid Med Cell Longev. 2019. PMID:31827702.

-------------------------------------------------------------------

While we have been doing various experiments using Jurkat T cell clones (JT/Neo, JT/BCL-2, J/Neo, and J/BCL-XL), we have not recognized any problem in these Jurkat T cell clones until now. Furthermore, BS-181-induced G1 cell cycle arrest and extrinsic apoptotic responses, which were observed in JT/Neo and JT/BCL-2, were also observed in Jurkat T cell clones (wild-type A3, FADD-deficient I2.1, and caspase-8 deficient I9.2). Please refer to the sentence in Page 10, Lines 273-275; “Under the same conditions, BS-181-caused G1 cell cycle arrest was not observed in A3 cells, but observed in I2.1 and I9.2 cells where the apoptotic cell death occurred to a significantly lesser extent compared to A3 cells.”

In particular, BS-181-induced G1 cell cycle arrest, which was observed in JT/BCL-2 cells but not in JT/Neo cells by flow cytometry, was also observed in J/BCL-XL cells but not in J/Neo cells (The data regarding J/Neo and J/BCL-XL cells have been included in the revision as Supplementary 1). Consequently, we are certain that the cellular responses observed in BS-181-treated Jurkat T cell clones employed in this study are reliable.

2) Sub-G1 in flow data (images) is questionable. Due to apoptosis and DNA fragmentation, Sub-G1 occurs for apoptosis, but this peak is somewhat close to G1, rather than Y-axis. Data here looks more like cell debris. Since BS-181 works in nanomolar (Introduction), uM levels were used by the authors, which may induce vast cell death in short time and production of vast cell debris. Consider to lower down the doses and repeat flow assays.

[Answer]

   I believe that by the Attune NxT flow cytometer (ThermoFisher Scientific, Waltham, MA, USA) used in this study is a very advanced analytical instrument. This flow cytometer has a program that can exclude the cell debris and doublets, which we could not see much in this study, from the analysis. In addition, as shown in Fig. 1c-f, the rates of BS-181-induced apoptotic sub-G1 cells are very similar to those of FITC-Annexin V-positive cells, indicating that the populations close to Y-axis should be small-sized apoptotic bodies containing fragmented nucleus but not just cell debris.

The IC50 value of BS-181 in nanomolar mentioned in the “Introduction Section” was obtained using a method of in vitro protein kinase activity assay. However, IC50 values of BS-181 measured against cell proliferation of tumor cell lines were in the ranges of 11.5‒30.5 μM, which are similar to our IC50 values in Table 1. In our hands, presumable, because of cell permeability of BS-181, its impacts causing G1 cell cycle arrest and extrinsic apoptotic pathway activation were not easily detectable when Jurkat T cells were treated with BS-181 alone at concentrations lower than 5.0 μM.

As I mentioned in the last paragraph of Discussion Section, BS-181 is an ideal drug due to its poor cell permeability. In this regard, ICE0942 as BS-181 analog that is know to retain the CDK7 selectivity of BS-181 has been developed and undergoing Phase I/II clinical trials.

3) It is also weird that to evaluate cell viability in 20 hours. It is usually in 48 or 72 hours with low doses (nanomolar in literature).

[Answer]

This comment is interesting. In my case, when I intend to elucidate cellular and molecular mechanisms underlying a drug-induced cell cycle arrest and/or apoptotic cell death, 24 h treatment condition is better than 48‒72 h treatment condition. That is because tumor cell lines employed as experimental models frequently cannot survive for a long time in the presence of some of validated biochemical inhibitors for caspases, protein kinases, and/or protein phosphatase, which are essential tools to treat to inhibit the cellular target proteins.

Furthermore, the viability of Jurkat T cells was not significantly affected by BS-181 at concentrations lower than 5.0 μM even after treatment for 72 h (data not shown). In addition, now we know that the IC50 value of ICE0942 as BS-181 analog against Jurkat T cell clones is 250‒500 nM upon 24-h treatment.

4) IC50 in JT/BCL-2 needed.

[Answer]

To meet this comment, IC50 values for JT/BCL-2 as well as J/Neo and J/BCL-XL cells were included.

5) "G1 arrest ONLY in JT/BCL2..." is questionable. G1 cell percentage decreased in JT/Noe cells due to apoptosis. Low doses in JT/Neo cells is necessary to test the G1 arrest.

[Answer]

Please referred to answer to the comment 1.

6) BS-181 induced extrinsic apoptosis through induction of receptors and ligands. It appear a novel mechanism of action. As a CDK7 inhibitor, how does it work? Hypothetical discussion may need.

[Answer]

I like this comment very much. I also would like to understand the mechanism responsible for BS-181-induced enhancements of cell surface TRAIL, DR4, DR5 levels in Jurkat T cells, which may be extremely complicate to understand. One more goal that I would like to elucidate is the mechanism underlying BCL-2 overexpression-caused prevention of extracellular secretion of TRAIL and DR4/DR5 proteins in Jurkat T cells.

At this moment, I do not even know that whether BS-181-induced these changes in Jurkat T cells are regulated either at transcription levels or at post-transcription levels. I hope that we will be able to answer my questions and thus the reviewer’s comment in near future.

7) Refs are completely de-formatted.

[Answer]

Based on the comment, the references have been retyped properly.

Round 2

Reviewer 3 Report

Authors should know the differences between the viability tests (different doses for longer treatment time) and death tests  for apoptosis and apoptotic proteins, etc. (higher doses for short time).